# Characterization of Collagen Structure in Normal, Wooden Breast and Spaghetti Meat Chicken Fillets by FTIR Microspectroscopy and Histology

**DOI:** 10.3390/foods10030548

**Published:** 2021-03-06

**Authors:** Karen Wahlstrøm Sanden, Ulrike Böcker, Ragni Ofstad, Mona Elisabeth Pedersen, Vibeke Høst, Nils Kristian Afseth, Sissel Beate Rønning, Nancy Pleshko

**Affiliations:** 1Nofima AS, Muninbakken 9-13, Breivika, 9019 Tromsø, Norway; Ulrike.Bocker@Nofima.no (U.B.); Ragni.Ofstad@Nofima.no (R.O.); Mona.Pedersen@Nofima.no (M.E.P.); Vibeke.Host@Nofima.no (V.H.); nils.kristian.afseth@nofima.no (N.K.A.); sissel.beate.ronning@nofima.no (S.B.R.); 2Department of Bioengineering, Temple University, Philadelphia, PA 19122, USA; npleshko@temple.edu

**Keywords:** chicken meat quality, muscle abnormalities, connective tissue, histology, FTIR microspectroscopy

## Abstract

Recently, two chicken breast fillet abnormalities, termed Wooden Breast (WB) and Spaghetti Meat (SM), have become a challenge for the chicken meat industry. The two abnormalities share some overlapping morphological features, including myofiber necrosis, intramuscular fat deposition, and collagen fibrosis, but display very different textural properties. WB has a hard, rigid surface, while the SM has a soft and stringy surface. Connective tissue is affected in both WB and SM, and accordingly, this study’s objective was to investigate the major component of connective tissue, collagen. The collagen structure was compared with normal (NO) fillets using histological methods and Fourier transform infrared (FTIR) microspectroscopy and imaging. The histology analysis demonstrated an increase in the amount of connective tissue in the chicken abnormalities, particularly in the perimysium. The WB displayed a mixture of thin and thick collagen fibers, whereas the collagen fibers in SM were thinner, fewer, and shorter. For both, the collagen fibers were oriented in multiple directions. The FTIR data showed that WB contained more β-sheets than the NO and the SM fillets, whereas SM fillets expressed the lowest mature collagen fibers. This insight into the molecular changes can help to explain the underlying causes of the abnormalities.

## 1. Introduction

Some decades ago, the poultry industry detected the first signs of abnormalities on chicken breast, i.e., *Pectoralis major* muscles [1]. One of these abnormalities, termed Wooden Breast (WB) [2], is characterized by fillets with a hard and bulged surface. Another less described abnormality is Spaghetti Meat (SM). SM fillets are soft with separated muscle fibers resulting in a stringy, spaghetti-like appearance [3]. These abnormalities make the meat appear unpleasant, resulting in the downgrading of WB and SM fillets. Both WB and SM abnormalities show similar morphological features in the muscles, such as fiber necrosis, inflammatory cell accumulation, and fibrosis [2,3,4]. Fibrosis occurs when the muscle fibers undergo necrosis, and the muscle fibers are replaced by connective tissue [5]. These abnormalities also have higher collagen and fat content in the superficial layer than the normal (NO) chicken and higher water, and lower protein content [3,6,7,8,9]. 

While these abnormalities’ muscle fibers are well-described [3,4,6,10,11], the connective tissue is less studied, particularly in SM. Connective tissue consists of a complex and heterogeneous matrix built up by collagen fibers, amorphous ground substances, and cells. The ground substance is formed mainly by glycosaminoglycans (GAGs) and structural glycoproteins linking the collagen fibers. Collagen fibers consist of tightly packed fibrils composed of three polypeptide chains, each in the form of a left-handed polyproline-type [12]. 

It has been suggested that the increased firmness in WB may be a result of fibrosis, which leads to an accumulation of interstitial connective tissue [9]. Chapman et al. (2017) suggested that the collagen content or maturity alone is not responsible for the increased stiffness of the fibrotic muscle in desmin knockout mice. The changes in collagen fiber organization in the perimysial layer caused the mechanical modifications [13,14]. In WB chicken, two types of collagen organization have been observed: (1) highly cross-linked collagen fibrils packed together to form larger collagen fibril bundles that are parallel aligned, and (2) thinner collagen fibril bundles that are more randomly arranged [4,15,16]. The collagen fibers in SM have not been characterized, even though fibrosis has been detected in SM. 

Fourier transform infrared (FTIR) spectroscopy has become a well-established tool to analyze complex biological sample components. By absorbing infrared light at specific wavenumbers, each chemical functional group contributes to a spectral fingerprint [17]. When combined with a microscope, FTIR analysis can quantify relative amounts, distribution, and orientation of compounds directly in biological tissues [18]. FTIR microspectroscopy and imaging have proven to be an excellent tool for analyzing protein structures in tissue related to microstructure [19]. The authors have previously shown that connective tissue components important for fish fillet quality were detectable with FTIR spectroscopy [20]. With Principal Component Analysis (PCA), it was possible to detect differences in the relative collagen content. The FTIR images revealed a collagen distribution that was corresponding to the collagen distribution illustrated by immunohistochemistry. In bone and skin, FTIR microspectroscopy and imaging have been used to assess collagen maturity related to mineralization and/or degree of mature crosslinks [21,22,23]. The orientation of collagen molecules is also an essential determinant of their functionality in connective tissues. Polarized FTIR spectroscopy can give structural information on oriented and ordered molecules and has been used to examine the molecular orientation of collagen fibers in cartilage [24] and skin [25]. In the current study, we hypothesize that the difference in texture properties among NO, WB, and SM chicken fillets can be explained by differences in the collagen fiber molecular structure and organization in the ECM. We characterized connective tissue focusing on collagen in chicken breast muscle with and without muscle abnormalities, using histology and FTIR spectroscopy. Using FTIR microspectroscopy, we obtained structural and chemical information of protein related to the proteins’ tertiary and secondary structures, while ratio images of amide I/amide II obtained by polarized FTIR revealed new information about the spatial orientation of the collagen fibers in the tissue. These data yield important insights related to the collagenous micro- and ultrastructure in chicken with breast abnormalities.

## 2. Materials and Methods

### 2.1. Chicken Fillets 

A total of 30 skin and boneless chicken breast fillets (*M. pectoralis major*) were obtained from a Norwegian commercial chicken processing facility. The birds of the strain Ross 308 were hatched 32 days old. Fillets were classified on-site as NO, WB, and SM by visual inspection by an experienced veterinarian based on palpation of consistency (normal, soft, hard). Breast fillets with soft consistency were classified as SM, breast fillets with hard consistency were classified as WB. The experiment consists of ten fillets from each chicken group.

### 2.2. Thin Section Sample Preparation 

Just after slaughter, cooling, and deboning, the chicken fillets were brought to the laboratory under refrigerated conditions (1 h). Approximately one cm^3^ from the upper part of each *Pectoralis major* muscle was cut out from each of the 30 fillets and fixed in 10% formalin buffer (for histology and FTIR analysis. The samples were kept for 24 h at room temperature, dehydrated in a graded series of ethanol, and embedded in paraffin. From each sample, a series of parallel sections, 5 µm in thickness, were cut and mounted on polylysine-coated slides for histological evaluation and on a ZnSe slides for FTIR spectroscopy. The sections were cut both transversally and longitudinally to the muscle length. 

### 2.3. Histology

For microscopic observations, three histological staining methods were used; fluorescently tagged wheat germ agglutinin (WGA) lectin was used to display an overview of the connective tissue by fluorescence microscopy, hematoxylin and erythrosine (HE) was used to verify the muscle abnormalities, and Picrosirius Red to give detailed information of the collagen structure. WGA binds sialic acid/N-acetylglucosamine sites in skeletal muscle tissue, whereas Picrosirius Red staining binds specifically to the collagen triple helix structure, thereby visualizing the collagen fibers in the tissue. 

The staining was done on deparaffinized sections, and the process is 2 × 5 min in xylene, rehydration in a series of ethanol (2 × 100; 2 × 95; 1 × 90; 1 × 70%) and rinsing with dH_2_O. One section from each of the 30 samples, i.e., ten sections from each fillet group (NO, WB, and SM) were stained with HE (Fisher, Fair Lawn, NJ, USA), dehydrated in the alcohol series back to xylene, and mounted in quick hardening mounting medium Eukitt^®^ (Merck, Darmstadt, Germany). The slides were scanned with Aperio CS 2. Based on the HE images, five fillets from each chicken group that had the typical morphological features described for NO, WB, and SM muscles, respectively, were selected for further analyses. For assessment of connective tissue, sections were permeabilized with 0.1% Triton X-100 in PBS for 15 min, incubated with WGA Alexa Fluor™ 488 Conjugate (Thermo Fisher Scientific, MA, USA) for 30 min, washed 3 × 10 min with PBS before using Dako fluorescent mounting medium (Glostrup, Denmark). The sections were examined by fluorescence microscopy (Zeiss Axio Observer Z1 microscope), and images were processed using Adobe Photoshop C3S. If necessary, the adjustment in brightness and contrast was performed manually across the entire image. For the collagen structure assessment [26], the sections were stained with Picrosirius Red Stain Kit (Polysciences, Warrington, PA, USA) according to the manufacturer’s protocol. The sections were then examined using a light microscope (Leica DM60001, Heidelberg, Germany). The images were taken in both bright-field and polarization mode. Using polarization mode, thick collagen fibers appear red and thin fibers appear green [26,27]. Picrosirius Red staining was done on both transversal and longitudinal cut sections. 

### 2.4. FTIR Micro Spectroscopy

The FTIR microscopy spectra were acquired with a Perkin Elmer Spectrum Spotlight 400 FTIR system (Perkin–Elmer, Buckinghamshire, UK). For the spectral analyses, the point function was used. The size of each point was 13 × 41 µm. All spectra were collected in transmission mode in the mid-infrared region between 4000–750 cm^−1^ with 32 scans per pixel and a spectral resolution of 4 cm^−1^. Before each spectrum, a background spectrum of the ZnSe was obtained. 

FTIR spectra were recorded on transversally cut sections on three different perimysium areas on each section. Five spectra were obtained along the connective tissue string in each of the three perimysium areas. This resulted in 15 spectra from each fillet. Five fillets from each chicken group were analyzed, resulting in 225 spectra for the three chicken groups altogether. For further analysis, five spectra from each of the three areas were averaged, and the mean spectra were used for peak height determination and principal component analysis.

### 2.5. Polarized FTIR Imaging

Polarized FTIR images were obtained with a Perkin Elmer Spectrum Spotlight 400 FTIR system (Perkin–Elmer, Buckinghamshire, UK) with a polarizer inserted in the light path. Polarization images were obtained on longitudinal sections aligned along the direction of the Y-axis of the motorized plate. The size of the IR spectral images was 75 × 125 µm with 6.25 µm pixel resolution. The data were recorded from 4000–750 cm^−1^ with a spectral resolution of 4 cm^−1^ and 120 scans per pixel. Before each image, a background spectrum with 240 scans was obtained from the ZnSe substrate. To determine the collagen fibers’ orientation, the angle of the polarization was set to 0°, which means that the polarization was perpendicular to the connective fiber length axis. Eight polarized images were collected from each of the groups, NO, WB, and SM. 

Spectral images were created using the amide I/amide II area ratio for the conventional mode and the perpendicular polarization mode on longitudinally cut muscle sections. The spectra were recorded in the perimysial area. The data of the amide I/amide II ratio were grouped into three categories: value ≤ 1.7 was considered valid for collagen fibers that were aligned parallel to the muscle fibers, a ratio between 1.7–2.7 for a random alignment of the collagen fibers, and for ratio value ≥ 2.7 the collagen fibers were considered to be perpendicular to the muscle fiber. This is according to the method used to determine collagen fibers’ alignment in cartilage [24]. The color scale was adjusted from 1.5 (blue) to 3 (red) according to the values computed for the three chicken groups’ connective tissue.

### 2.6. Spectral Data Analysis

The main spectral regions of interest for proteins are the amide I and II region (1718–1492 cm^−1^) of the FTIR spectra, the amide III region (1350–1200 cm^−1^), and the region 1140–985 cm^−1^ for the carbohydrate-rich matrix components (proteoglycans (PG), GAG). Before further processing the spectral data, the absorbance of paraffin in the FTIR spectra were deleted from the spectra [28], i.e., the areas between 1500–1440 and 1400–1360 cm^−1^ were removed. Additionally, to increase the resolution of the peaks underlying absorbance bands, a second derivative algorithm was applied (Savitzky & Golay, 1964). The window size was nine smoothing points, and the polynomial order was two. For normalization, extended multiplicative signal correction (EMSC) was applied [29] for the whole spectral region (4000–750 cm^−1^). After preprocessing, the data were analyzed by determining peak heights and Principal Component Analysis (PCA). In the second derivative spectra, peaks appear negative; to get positive values, the spectra were multiplied with (−1). The multivariate analysis PCA describes the maximum variance in the data set at all frequencies simultaneously. The results appear in a score plot that allows visualization of the distribution of the samples. The loadings explain the variance seen in the score plot [30]. 

Collagen maturity was calculated as the ratio of two sub-bands in the amide I contour at 1660 cm^−1^ and 1690 cm^−1^, as we previously described [22,23,31]. To calculate the relative content of GAGs to collagen in the connective tissue, the peak height ratio of the two bands at 1126 cm^−1^, assigned to the sulfate stretch in GAG [32,33], and 1660 cm^−1^, assigned to stretching vibration of the carbonyl in the triple-helix structure of collagen, was used. 

The FTIR polarized images amide I/amide II ratios were calculated by integrating areas and then rationing the areas using ISys^®^ software (Spectral Dimensions, Olney, MD, USA). All spectra were baseline corrected before obtaining the ratio, and spectra originating from muscle fibers were discarded by masking to avoid their influence on the connective tissue images.

A general linear model ANOVA followed by the post hoc Tukey test and 95% confidence were used to calculate differences in peak heights (1660/1690 cm^−1^ and 1126/1660 cm^−1^). The chicken was used as a random factor in the groups.

## 3. Results

The collagen organization within NO, WB, and SM chicken fillets was examined with microscopy and spectroscopic analysis. The histological images gave an overall view of connective tissue and muscle morphology, while FTIR spectroscopy provided a more detailed characterization of the chemical and molecular structure of collagen.

### 3.1. Histological Characterization of Skeletal Muscle Abnormalities

To verify the presence of WB and SM abnormalities in the selected fillets, histological evaluations with three different staining methods were performed on the fillets. WGA staining of the connective tissue showed that WB had visually clearly more connective tissue (Figure 1 upper level).

The HE-images (Figure 1 lower level) showed that the NO fillets displayed even-sized, tightly packed myofibers and that the perimysium was narrow. In the WB fillets, the muscle cells were round and swollen, with large diffuse, widened perimysial connective tissue areas. The perimysium in SM was as for the WB sample broad and diffuse, but the muscle cells were more tightly packed in the SM tissue. The morphology of NO, WB, and SM samples visualized by HE was in accordance with other studies [2,3,4]. The histological images confirmed that the selected chicken fillets were sorted correctly into the three classes NO, WB, and SM. 

Picrosirius Red staining and light microscopy analysis in bright-field or polarized light were used to characterize the collagen organization in the tissue. In bright-field mode (Figure 2 upper level), the perimysium in the NO samples appears as intense continuously red, and with tightly packed thread-like structures indicating that the collagen fibers are highly organized in thick bundles oriented transversally to the long axis of the muscle cells. 

WB and SM’s perimysium are both inhomogeneous with gaps and containing short red threads oriented in multiple directions. The WB has a denser matrix structure and more red threads than the SM sample, and the perimysium in WB seems to intrude into the muscle fiber bundles. The section background is black in polarized light, and the collagen fibers appear red (thick fibers) and green (thin fibers). The perimysium in the NO fillet appears as a compact and continuous red string (Figure 2 lower level), indicating that the collagen bundles are even-sized, parallelly aligned, and tightly packed [26]. The stained perimysium in WB and SM is much less coherent with a mixture of red and green colored short threads. The SM sample has visually inspected more green threads than the WB samples. This may indicate that the collagen fiber structure and organization in the connective tissue are different in the three chicken fillet groups. 

### 3.2. FTIR Microspectroscopy

FTIR spectra were collected from the perimysium of NO, WB, and SM fillets. EMSC corrected spectra representing the three groups are shown in Figure 3A. 

Here, the wavenumber regions between 1500–1440 and 1400–1360 cm^−1^ were deleted as the paraffin absorbance masks other protein and carbohydrate signals in these regions. The main peaks were found in the amide I and II regions (1718–1492 cm^−1^), a weaker and broader peak in the amide III region (1350–1200 cm^−1^), and a weak broad peak in the carbohydrate region (1140–985 cm^−1^). In connective tissue, collagens are the most extensive protein constituents. Therefore, the infrared absorbance in the protein area primarily originates from the absorbance of collagens. Figure 3B,C show the average second derivative spectra of the amide I and II region and the carbohydrate region, respectively. The most pronounced protein peaks (negative in the second derivative) were found at approximately 1690, 1660, and 1548 cm^−1^, with an additional shoulder at 1638 cm^−1^. The 1660 and 1548 cm^−1^ peaks have previously been assigned to native triple-helical structure in collagens [34]. The shoulder at 1638 cm^−1^ has been related to β-sheet structures [35]. The bands at 1084, 1062, and 1033 cm^−1^ are assigned to the sugar rings in carbohydrate residues in collagen and proteoglycans (PGs). Besides, there is a band at 1126 cm^−1,^ which has been assigned to the sulfate stretch in GAGs [32,33,36].

To better explain the collagen structure differences between the sample groups, the spectra were further compared using PCA. The score plot revealed only partly separation between the three groups (see supplementary Appendix A: PCA Score plot for PC1 and PC2 of the Amide I & II region). Thus, to improve interpretation, PCA of two and two groups were performed, respectively. The score plot of NO and SM showed no apparent differences between the samples (not included). In Figure 4A, the score plot reveals that PC 1 explains 44% of the SM and WB spectra variation. 

The loading plot shows that the difference between WB and SM is mainly related to a shift in the FTIR band frequency around 1660 cm^−1^, assigned to the native triple-helical structure of collagen. For SM samples, this band is slightly shifted to lower wavenumber (i.e., 1654 cm^−1^, assigned to α-helixes), indicating that the collagen in SM samples has a lower content of native triple-helical structure compared to WB samples [37]. The loading plot also shows that bands around 1638 (β-sheet structures) and 1126 cm^−1^ (sulfate stretch in GAGs) can explain differences between WB and SM. The PCA score plot of NO and WB (Figure 4B) shows differences between WB and NO samples along with PC 2, explaining 21% of the variance. The loading plot indicates that this difference is related to the bands at approximately 1660 and 1548 cm^−1^, meaning that NO samples have relatively higher intensities of native triple helix than WB samples. 

Figure 5 shows the absorbance ratio of 1660/1690 cm^−1^, which indicates that the NO fillets, followed by WB fillets, have the most mature collagen fiber structures. 

The SM fillets express the lowest degree of maturation, and it is significantly different from the NO fillets. WB was not significantly different from either the NO or SM fillets. The peak height ratio of 1126 cm^−1^, assigned to the sulfate stretch in GAG, and 1660 cm^−1^, assigned to the collagen’s triple-helix structure, is shown in Figure 5. This ratio indicates that the SM samples have significantly more GAGs relative to collagen in the perimysial connective tissue than both the NO and the WB samples. The standard deviation is more extensive in the SM sample than in the two other samples, indicating a sizeable focal variation in the connective tissue structure in the SM samples.

### 3.3. Polarized FTIR Imaging

Parallel longitudinally sections were stained with Picrosirius Red and analyzed in bright-field light microscopy for comparison with polarized FTIR images (Figure 6).

The bright-field images show that the NO has a dense connective tissue with visible collagen fibers. WB and SM have more diffuse and broader connective tissues with more gaps. 

The primary absorbance in the amide I is the stretching vibration of the C=O band (perpendicular to the collagen fibril axis). The C-N stretching and N-H bending vibration (parallel to the collagen fibril axis) contribute to the amide II band. In the polarized FTIR images (Figure 6), low amide I/amide II ratio values represent collagen fibers aligned parallelly, whereas higher values represent an unordered collagen fiber structure having multiple directions. The blue-green pixel color of the perimysial connective tissues in the NO samples indicates that the collagen fibers are mainly parallel aligned to the muscle fiber axis. The red-yellow pixel colors in both the WB and the SM samples show that the collagen fibers in these samples compared to the NO samples have a higher amide I/II ratio, indicating that they are more randomly oriented. The average amide I/II ratio values are 2.05, 2.39, and 2.41 for NO, WB, and SM, respectively. 

## 4. Discussion

Differences in meat quality and textural properties between normal chicken and those with muscle abnormalities have been related to muscle fiber necrosis [38]. More recently, differences in the amount, structure, and ECM composition have been reported [2,3,4].

Both the HE- and the Picrosirius Red stained images revealed that the perimysium of NO fillets had a dense appearance. In contrast, for the WB and SM samples, the perimysium appeared transparent and inhomogeneous with some dense stained collagen fibrils area and gaps. In the bright-field mode images, Picrosirius Red staining was more intense in NO than in WB and least severe in SM. Baldi et al. (2019) [39] reported that WB samples had higher collagen content than both the NO and the SM samples when measured chemically as collagen content per mg muscle tissue. The higher collagen content of WB fillets than in the NO, as reported by Baldi et al. (2019) [39], probably is because there is more connective tissue distributed in the superficial layer of the breast muscle in WB than in NO and SM samples, which is in line with our histological data. The samples’ low magnification WGA images clearly showed that the connective tissue was more abundantly distributed within the muscles of WB. Using polarized light in combination with Picrosirius Red staining, thick collagen fibers appeared red and thin collagen fibers green [26,27]. The polarized light images revealed that in the NO perimysium, the collagen fibers are thicker than in the WB perimysium, having a mixture of thick and thin fibers and that the SM perimysium had mainly thin collagen fibers. The mechanical properties of collagen fibers are dependent on their diameter and degree of maturity, which could impact the muscle’s texture.

In the FTIR spectra of connective tissue, the main contributor to the amide bands was collagen. Since the spectra were normalized, the absorbance differences mainly reflect non-quantitative differences in the protein structures between the samples. However, if there are large quantitative differences, these will also be reflected in the absorbance spectra and be enhanced in the second derivative spectra. The second derivative spectra revealed differences in the collagen structure and abundance between WB and the two other samples related to the bands at approximately 1690, 1660, 1638, and 1548 cm^−1^. The NO and SM sample groups showed the highest absorbance at 1660 and 1548 cm^−1^, indicating a higher amount of native triple helical structures. A similar decrease in the 1660 cm^−1^ band component was reported by Payne & Vies (1988) while investigating denaturation of the collagen triple helix in collagen extracted from lathyritic rat [31,34]. When comparing WB with SM and NO separately by PCA, the WB fillets differed from the SM by a shift in 1660 cm^−1^ peak. The SM fillets shifted towards a lower wavenumber (1654 cm^−1^). An α-helix has an absorbance at 1654 cm^−1^_,_ while a native triple helix has an absorbance at 1660 cm^−1^; this can mean that the SM fillets consist of more loosely bound α-helixes and that WB fillets have a more triple helical structure where the collagen fibrils are tightly packed together. The PCA of NO and WB indicated that the differences between them are 1660 and 1548 cm^−1^, meaning that the NO fillets have more tightly packed triple-helical than WB. The FTIR absorbance peak at 1126 cm^−1^, assigned to GAGs, is most pronounced in the SM fillets. This is in accordance with the more amorphous appearance of the connective tissue in the microscopic images. This ratio’s large standard deviation may indicate that the degree of fibrosis varies locally within the muscle.

The absorbance ratio of 1660/1690 cm^−^^1^ is thought to be related to determining the degree of maturity of collagen in bone and skin [22,23,40]. Canuto et al., and Paschalis et al., ascribe the higher ratio of 1660/1690 cm^−1^ to a higher number of non-reducible cross-links in skin and bone, respectively. Farlay et al. [31], on the other hand, relate an increase in the band ratio of 1660/1690 cm^−1^ rather to alterations in mineralization and hydration leading to changes in collagen secondary structure of bovine bone. This ratio has previously not been validated as a marker of collagen maturity in the connective tissue of skeletal muscle. Nevertheless, it is plausible that its interpretation concerning collagen maturity in the sense of changes in collagen’s secondary structure would be similar. Immature, reducible cross-links consist of two procollagens that are in a parallel alignment. A mature, non-reducible cross-link can only be formed through an additional collagen molecule aligned to the divalent cross-linked molecule, which means that a mature cross-link exists when three or more collagen fibrils are cross-linked to each other [21]. In a recent study, Baldi et al. (2019) [39] showed that the intramuscular connective tissue in SM’s superficial muscle contains less hydroxylysylpyridinoline per mol collagen than NO and WB muscles. SM fillets expressed the lowest degree of maturity in the present study, measured as the ratio 1660/1690 cm^−1^. This was supported by the Picrosirius Red staining seen in the polarized microscopic images. The collagen fibers in SM appeared as thin green colored threads and poorly packed, consisting of more loosely bound α-helixes (1654 cm^−1^). In our study, the absorbance ratio of 1660/1690 cm^−1^ was highest for the NO fillets showing the highest degree of maturity, closely followed by the WB fillets. The color staining in the microscopic images confirmed the presence of thicker collagen fibers in the NO and partly in the WB samples. If SM consists of newly formed collagen fibers, the tissue most likely consists of immature divalent cross-links that will convert into mature trivalent cross-links during maturation. In the present study, the results indicated that SM abnormality exhibited significantly lower collagen cross-linking than NO fillets. The absorbance ratio of 1660/1690 cm^−1^ reflects the content of mature and immature collagen in the connective tissue. 

In WB muscle, the perimysial connective tissue contains more collagen, a higher amount of thicker collagen fibers, and the connective tissue appears denser than SM’s perimysium. The connective tissue in WB is likely a mixture of thin collagen fibers and thicker mature collagen fibers. More collagen fibrils are produced through fibrosis, and the connective tissue converts into a stiffer, more mature cross-linked dense collagen structure, causing the hard-bulged surface of WB fillets. WB differs from the two other sample groups by having a slightly more pronounced shoulder at 1638 cm^−1^, assigned to β-sheets structures. Perez-Puyana et al. (2019) [41] found when comparing four different collagen and gelatin concentrates that a higher amount of β-strands went along with a higher degree of denaturation. The β-sheet structures are generally more stable than the helix structures. Proteins that contain large fractions of β-sheets usually exhibit higher denaturation temperatures. Baldi et al. (2019) reported that the stromal protein fraction (collagen, elastin, reticulin) in WB meat taken from the superficial layer had higher denaturation enthalpy than NO and SM, as well as samples from the inner muscle of WB. The higher content of β-sheets structures may contribute to a stiffer structure. 

Polarized FTIR-spectroscopy confirmed that NO samples displayed a well-ordered structure, while WB and SM samples showed more randomly organized collagen fibers. This is in accordance with previous findings in chicken with abnormalities [4,15,16]. Bi et al. (2005) have shown that low values of amide I/II indicated ordered structure and fiber alignments, whereas a high ratio was related to unordered structures in cartilage [24]. This quantitative interpretation seems to be true also for connective tissue.

We hypothesize that the difference in texture properties between NO, WB and SM chicken fillets can be explained by differences in the collagen fiber molecular structure and connective tissue organization. Both FTIR spectroscopic and histological results show that the collagen structure in the perimysium in NO fillets consists of mature thick collagen fiber bundles, tightly packed in well-organized structures. In WB, the connective tissue is more abundant but consists of a mixture of thin and thick collagen fibers that are randomly organized. The more rigid texture of WB compared to NO may thus be explained mainly by more extensive deposition of connective tissue in the former. The SM abnormality has thinner and looser, immature collagen fiber bundles that are randomly organized and more of the ground substance. This structure can explain the connective tissue’s fragility, causing the muscle fibers to be teared apart during handling and filleting. The differences in the connective tissue may explain the textural differences between WB and SM. It cannot be overlooked that the textural differences are related to the fibrosis and the chemical and structural states of the meat proteins [42]. Together, these data yield new insight into the molecular changes that underly connective tissue abnormalities in the Wooden Breast and Spaghetti Meat syndromes. In the past few years, several studies have tried to elucidate the underlying mechanisms of chicken myopathy, using either gene expression analysis, proteomics or standard histological assessment, however the exact etiology and the chronology of events leading to developing this condition is still only partially understood [43,44]. Our finding reveals a disorganized micro-and macro organization of collagen structure to be a part of this complex pathogenesis. Further studies focusing on collagen interacting molecules and regulators of fibrillogenesis could be an important contribution toward a complete ethology of this chicken myopathy.

## Figures and Tables

**Figure 1 foods-10-00548-f001:**
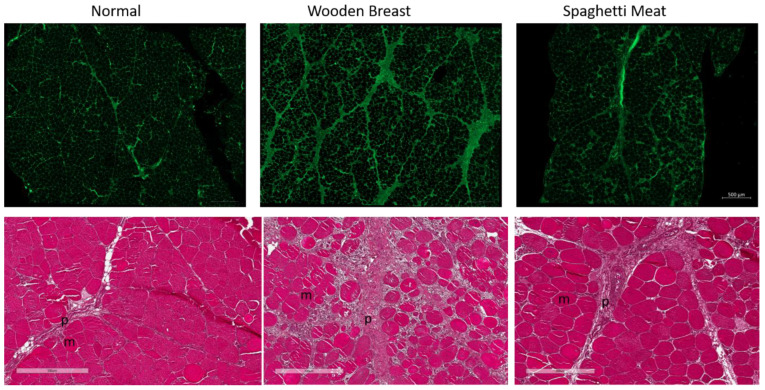
Upper level: Fluorescent WGA lectin staining of muscle sections to visualize connective tissue distribution in Normal, Wooden Breast and Spaghetti Meat muscles. Scale bar is 500 µm. Lower level: HE-stained tissue sections collected from Normal, Wooden Breast, and Spaghetti Meat chicken breast muscle. m = muscle cell, p = perimysium. Scale bar is 300 µm.

**Figure 2 foods-10-00548-f002:**
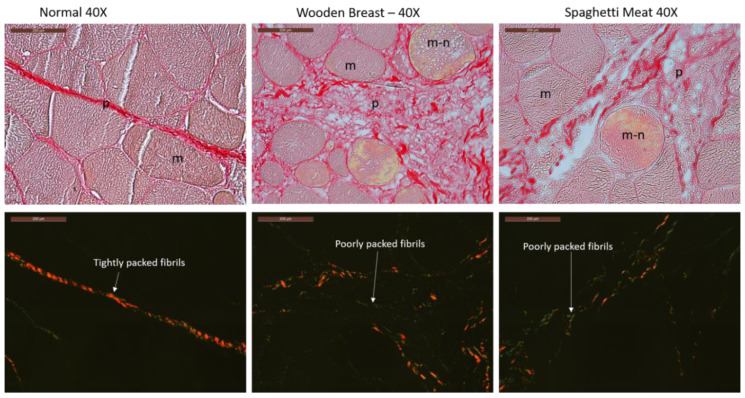
Picrosirius red staining: Bright-field mode (**upper panel**) and polarized light microscopy (**lower panel**). Thick collagen fibrils appear as red while thin fibrils as green. m = muscle cell, m-n = muscle cell infiltrated with macrophages and undergoes necrosis p = perimysium, e = endomysium. Scale bar: 200µm.

**Figure 3 foods-10-00548-f003:**
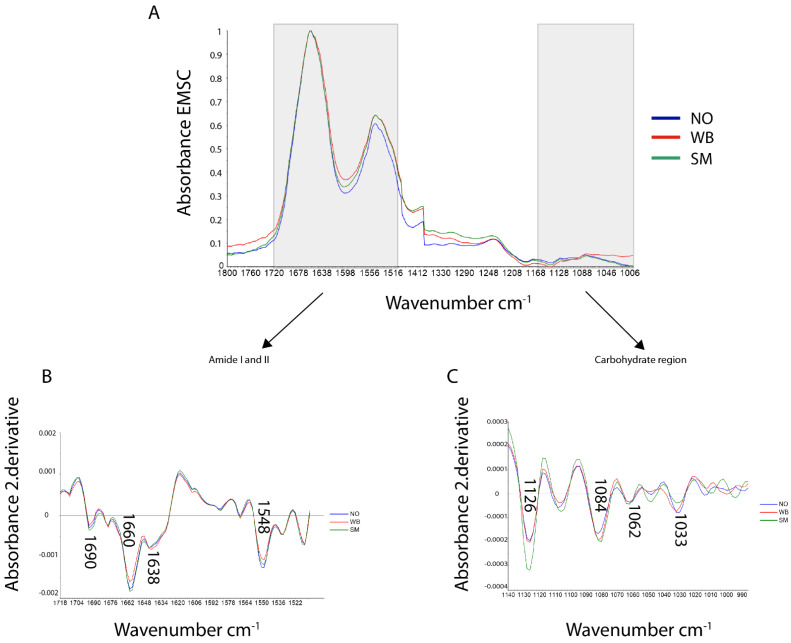
(**A**) FTIR spectra (averaged and EMSC corrected) collected from the connective tissue in NO, WB, and SM chicken fillet (1800–1000 cm^−1^) (**B**) Savitzky Golay’s second derivative spectra of NO, WB, and SM fillets in the Amide I & II region (1718–1492 cm^−1^) and (**C**) carbohydrate region (1140–985 cm^−1^).

**Figure 4 foods-10-00548-f004:**
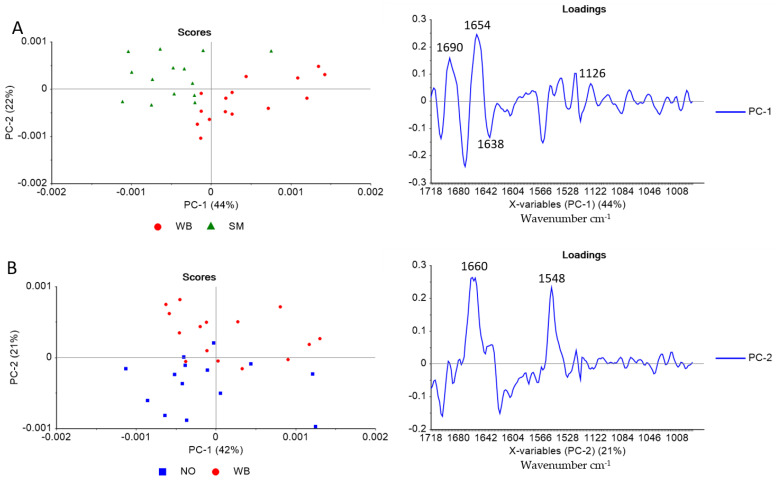
PCA Score and loading plot for the amide I & II region (1718–1492 cm^−1^) and the carbohydrate region (1140–985 cm^−1^) for (**A**) WB and SM samples; and (**B**) NO and WB samples.

**Figure 5 foods-10-00548-f005:**
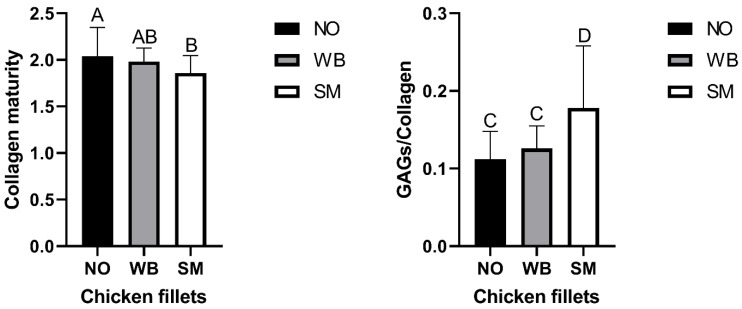
(**Left**): Collagen maturity (1660 cm^−1^/1690 cm^−1^): The NO and WB fillets show the most mature collagen network, the SM fillets express the lowest degree of maturation, and it is significantly different from NO (*p* < 0.05). (**Right**): GAGs to collagen ratio (1126 cm^−1^/1660 cm^−1^): SM has a significantly higher content of GAGs than NO and WB. Different letters mean a significant difference for *p* < 0.05.

**Figure 6 foods-10-00548-f006:**
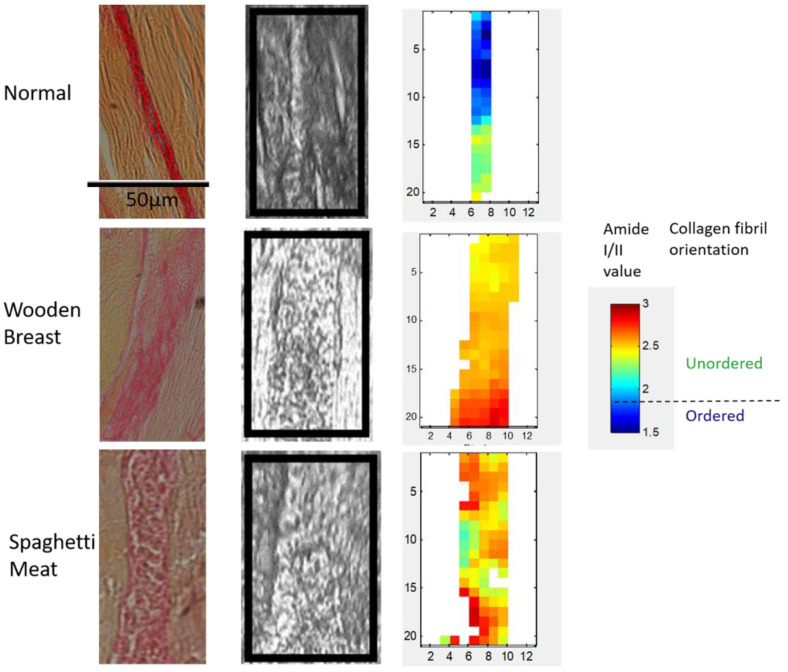
(**Left**): a picrosirius red-stained image of a longitudinal section of NO, WB, and SM connective tissue, (**middle**): light microscopy image, and (**right**): the corresponding polarized FTIR images calculated from the ratio amide I/amide II. Low ratios values (blue-green color) indicate more ordered fibers and high values (yellow-red) less ordered fiber organization.

## Data Availability

Data available on request due to restrictions.

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
