# Peer review of "Characterization of Collagen Structure in Normal, Wooden Breast and Spaghetti Meat Chicken Fillets by FTIR Microspectroscopy and Histology"

_foods, 2021, doi:10.3390/foods10030548_

Round 1
Reviewer 1 Report
This paper reports a study about the comparison of the properties of collagen obtained from different sources. In this line, collagen from Normal, Wooden Breast and Spaghetti Meat chicken fillets has been analysed through FTIR. The paper includes interesting results with suitable experimental design and data analysis. In this sense, after major revision indicated below.
FORMAT
- Revise the format of the References Section following the Guide for Authors from the journal.
GRAMMAR
- English language and style are fine.
FIGURES
- Improve the quality of Figures 3 and 5.
- Include the legends for the axis of the graphs in Figures 3 and 5.
MATERIALS
- Include a Materials Section (normally 2.1) to include other reagents used in the study.
METHODS
- All the methods have been properly described.
DISCUSSION
- Improve the discussion of the results from the FTIR spectra. Compare them with other author’s results:
- https://doi.org/10.1515/hsz-2019-0206
- https://doi.org/10.1016/j.foodchem.2003.09.038
- https://doi.org/10.1002/bip.360271105
REFERENCES
- Include more references from the journal.
Specific Comments
- Include possible further studies related to this work that can be carried out.
- Include the main novelty of the study.
- “Collagen fibers consist of tightly packed fibrils composed of three polypeptide chains, each in the forma of a left-handed polyproline-type helix” (Lines 43-44). Include a reference to support this statement.
Author Response
Dear reviewer,
Thank you for revision of my manuscript and for suggestions and comments. We considered the comments with care and acted on the majority of the points raised by the reviewer. Below, please find our answers to your comments (red text) presented in the same order as in the review.
Yours sincerely,
Karen Wahlstrøm Sanden
FORMAT
- Revise the format of the References Section following the Guide for Authors from the journal.
The endnote template (MDPIcopy) suggested by the journal is used. The references that were incomplete have been corrected.
FIGURES
- Improve the quality of Figures 3 and 5.
- Include the legends for the axis of the graphs in Figures 3 and 5.
Figures 3 and 5 have been improved and a legend on the axis is now included.
MATERIALS
- Include a Materials Section (normally 2.1) to include other reagents used in the study.
All chemical reagents used in this study are listened in the respective method sections where they naturally belong. For instance, we believe that chemicals used during histology should not be removed from the corresponding method description, since they are a core part of these procedures (this is also common practice).
DISCUSSION
- Improve the discussion of the results from the FTIR spectra. Compare them with other author’s results:
- https://doi.org/10.1515/hsz-2019-0206
- https://doi.org/10.1016/j.foodchem.2003.09.038
- https://doi.org/10.1002/bip.360271105
Two of suggested references are now included as part of the discussion as seen in line 358-360 and 407-409.
REFERENCES
- Include more references from the journal.
https://doi.org/10.3390/foods10010104 is now included to in the discussion line 433-436. Additionally we also included doi:10.1038/nrg3185 to this section.
Specific Comments
- Include possible further studies related to this work that can be carried out.
There is a section on possible further studies added in line 437-446.
- Include the main novelty of the study.
The novelty of this study is now pointed out in line 80-82.
- “Collagen fibers consist of tightly packed fibrils composed of three polypeptide chains, each in the forma of a left-handed polyproline-type helix” (Lines 43-44). Include a reference to support this statement.
A reference is now added in line 44.
Reviewer 2 Report
The present study reports a nicely developed study and is of interest to the readers of foods.
The manuscript is generally well written, and the data are presented with quality and adequately discussed.
My major concern has to do with the low number of samples analysed. 30? 10 per groups? In this kind of FTIR studies, 300, namely 100 per group seems more reasonable number! If the authors have already validated their model, please specifically make a reference to it in the manuscripts with cited references.
Author Response
Dear reviewer,
Thank you for revision of my manuscript and for comments. Below, please find our answer to your comment (red text).
Yours sincerely,
Karen Wahlstrøm Sanden
The present study reports a nicely developed study and is of interest to the readers of foods.
The manuscript is generally well written, and the data are presented with quality and adequately discussed.
My major concern has to do with the low number of samples analysed. 30? 10 per groups? In this kind of FTIR studies, 300, namely 100 per group seems more reasonable number! If the authors have already validated their model, please specifically make a reference to it in the manuscripts with cited references.
The reviewer raises an interesting point. The objective in this study was to gain more insight in detailed structural differences between the sample groups. These three groups were selected due to clearly visible gross characteristics by expert eye and should therefore by highly representative samples. Increasing the number of samples 10-fold would be extremely resource-demanding due to the nature of the histological methods as would be analyzing several hundred samples section by FTIR microspectroscopy and polarized FTIR. In the present study, the spectroscopic methods used are not aiming at e.g. calibrating a model to be used for classification purposes, where a larger number of samples, collected over a longer period of time would be necessary. However, this will be needed for future validation of the approach, which could be a natural follow-up of the present study. We have added a couple of sentences in the discussion to clarify this point.
Reviewer 3 Report
The manuscript intitled “Characterization of collagen structure in normal, wooden breast and spaghetti meat chicken fillets by FTIR microspectroscopy and histology” aimed to characterize the structure changes in defective chicken meat by histochemistry, polarized FT-IR microspectroscopy and polarized microscopy. Authotrs demonstrated a higher content in perimysium connective tissue and changes in collagen fibers structure in some abnormal meats compared to control ones.
The experimental design is good, the methods well chosen to meet the scientific objective and the manuscript is particularly well written, detailed and clear. The results are clearly presented and very well discussed in a specific section.
Thus the manuscript is very good but there are some errors in the reference list that must be corrected.
Minor comments:
In the section Material and Methods related to spectral data analysis:
Did the EMSC was calculated refering to the whole spetra (4000-750 cm-1 exluding absorption bands of paraffin) and from the whole dataset, or refering to “parts” of the whole spectra (1800-1000 cm-1; 1718-1492 cm-1; 1140-985 cm-1) ?
Figures
Lines 227 and 261: typo, remove a point at the end of the caption of Figure 2 and Figure 4.
Lines 242 and 267: Add the units on the figure 3 axes and on the x-axis of PCA loadings of figure 4 (cm-1)
Lines 284-286: Specify in the legend that different letters means a significant difference for p<0.05 (and change the legend to avoid repats)
References
Lines 449, 464: references of Velleman et al. incomplete (journal ?)
Lines 472, 476, 478, 486, 498, 509, 511, 518, 527, 529 : incomplete references (journal ?)
Author Response
Dear reviewer,
Thank you for revision of my manuscript and for suggestions and comments. We considered the comments with care and acted on the points raised by the reviewer. Below, please find our answers to your comments (red text).
Yours sincerely,
Karen Wahlstrøm Sanden
Comments and Suggestions for Authors
The manuscript intitled “Characterization of collagen structure in normal, wooden breast and spaghetti meat chicken fillets by FTIR microspectroscopy and histology” aimed to characterize the structure changes in defective chicken meat by histochemistry, polarized FT-IR microspectroscopy and polarized microscopy. Authotrs demonstrated a higher content in perimysium connective tissue and changes in collagen fibers structure in some abnormal meats compared to control ones.
The experimental design is good, the methods well chosen to meet the scientific objective and the manuscript is particularly well written, detailed and clear. The results are clearly presented and very well discussed in a specific section.
- Thus the manuscript is very good but there are some errors in the reference list that must be corrected.
The reference list is corrected.
Minor comments:
In the section Material and Methods related to spectral data analysis:
- Did the EMSC was calculated refering to the whole spetra (4000-750 cm-1 exluding absorption bands of paraffin) and from the whole dataset, or refering to “parts” of the whole spectra (1800-1000 cm-1; 1718-1492 cm-1; 1140-985 cm-1) ?
EMSC applied calculated on the whole spectral region after the paraffin absorbance was removed. This is now specified in the Materials and Methods line 178/179.
Figures
- Lines 227 and 261: typo, remove a point at the end of the caption of Figure 2 and Figure 4.
The points are now removed.
- Lines 242 and 267: Add the units on the figure 3 axes and on the x-axis of PCA loadings of figure 4 (cm-1)
Units have now been added.
- Lines 284-286: Specify in the legend that different letters means a significant difference for p<0.05 (and change the legend to avoid repats)
The legend has now been specified. The letters have been changed.
References
- Lines 449, 464: references of Velleman et al. incomplete (journal ?)
The references have been corrected.
- Lines 472, 476, 478, 486, 498, 509, 511, 518, 527, 529 : incomplete references (journal ?)
The references have been corrected.
Round 2
Reviewer 1 Report
Authors performed all the changes suggested in the previous review process. Therefore, this manuscript is recommended for publication.